# An Acetylcholinesterase Inhibition-Based Biosensor for Aflatoxin B_1_ Detection Using Sodium Alginate as an Immobilization Matrix

**DOI:** 10.3390/toxins12030173

**Published:** 2020-03-11

**Authors:** Amani Chrouda, Khouala Zinoubi, Raya Soltane, Noof Alzahrani, Gamal Osman, Youssef O. Al-Ghamdi, Sameer Qari, Albandary Al Mahri, Faisal K. Algethami, Hatem Majdoub, Nicole Jaffrezic Renault

**Affiliations:** 1Department of Chemistry, College of Science Al-Zulfi, Majmaah University, Al-Majmaah 11952, Saudi Arabia; amain.c@mu.edu.sa (A.C.); yo.alghamdi@mu.edu.sa (Y.O.A.-G.); 2Laboratory of Interfaces and Advanced Materials, Faculty of Sciences, Monastir University, Monastir 5000, Tunisia; zinoubikhaoula@yahoo.fr (K.Z.); hatemmajdoub.fsm@gmail.com (H.M.); 3Institute of Analytical Sciences, UMR CNRS-UCBL-ENS 5280, 5 Rue la Doua, 69100 Villeurbanne, CEDEX, France; nicole.jaffrezic@univ-lyon1.fr; 4Faculty of Sciences of Tunis, Tunis El Manar University, El Manar, Tunis 1068, Tunis; Rasoltan@uqu.edu.sa; 5Department of Basic Sciences, Adham University College, Umm Al-Qura University, Adham 21971, Saudi Arabia; neszahrani@uqu.edu.sa; 6Department of Biology, Faculty of Applied Sciences, Umm Al-Qura University, Makkah 21955, Saudi Arabia; 7Research Laboratories Center, Faculty of Applied Science, Umm Al-Qura University, Mecca 21955, Saudi Arabia; 8Agricultural Genetic Engineering Research Institute (AGERI), ARC, Giza 12915, Egypt; 9Department of Biology, Aljumum University College, Umm Al-Qura University, Makkah Aljumum 21955, Saudi Arabia; shqari@uqu.edu.sa; 10Department of Biology, Faculty of Sciences and Arts, King Khalid University, Dhahran Aljanoub 61421, Saudi Arabia; almohry@kku.edu.sa; 11Department of Chemistry, College of Science, Imam Mohammad Ibn Saud Islamic University (IMSIU), Riyadh 11432, Saudi Arabia; falgethami@imamu.edu.sa

**Keywords:** biosensor, acetylcholinesterase, aflatoxin B 1, sodium alginate, biopolymer

## Abstract

In this study, we investigated a novel aflatoxin biosensor based on acetylcholinesterase (AChE) inhibition by aflatoxin B1 (AFB1) and developed electrochemical biosensors based on a sodium alginate biopolymer as a new matrix for acetylcholinesterase immobilization. Electrochemical impedance spectroscopy was performed as a convenient transduction method to evaluate the AChE activity through the oxidation of the metabolic product, thiocholine. Satisfactory analytical performances in terms of high sensitivity, good repeatability, and long-term storage stability were obtained with a linear dynamic range from 0.1 to 100 ng/mL and a low detection limit of 0.1 ng/mL, which is below the recommended level of AFB1 (2 µg/L). The suitability of the proposed method was evaluated using the samples of rice supplemented with AFB1 (0.5 ng/mL). The selectivity of the AChE-biosensor for aflatoxins relative to other sets of toxic substances (OTA, AFM 1) was also investigated.

## 1. Introduction

Aflatoxin B1 (AFB1), as a mycotoxin, is one of the most toxic natural products as it is a common contaminant of human food and animal feed [1]. AFB1 exhibits mutagenic and teratogenic effects and also causes human hepatic and extrahepatic carcinogenesis [2]. AFB1 is known to occur naturally in agricultural products such as peanuts, corn, and animal feeds. These contaminated food or animal feeds present serious health hazards [3]. In order to ensure human and animal health, many countries have established its maximum permissible regulatory and occurrence level [4]. The limitations for AFB1 defined by the Tunisian National Standard (TNS 1983) and the European Commission in animal feeds and entire foodstuffs for dairy animals are 20 and 5 µg/kg, respectively (European Commission 2003) in corn, groundnuts, nuts, dried fruit, and cereals for human food [5]. Thus, due to the low permissible limit and severe toxicity of AFB1, developing rapid, sensitive, and specific analytical methods for its detection is vital. The widely used methods to identify toxic substances in the food industry include thin-layer chromatography (TLC), gas chromatography-mass spectrometry (GC/MS), high-performance liquid chromatography (HPLC), capillary electrophoresis (CE), and a variety of immunoassays [6]. Among these, biosensing approaches are potential substitutes for the recognition of aflatoxins, with the major advantages being their high sensitivity and specificity, cost-effectiveness, speed, and portability [7]. Acetylcholinesterase (AChE), an important enzyme in the transmission of nerve signals has recently become the most frequently used enzyme in biosensors due to its sensitivity to numerous toxic substances, pesticides, and glycoalkaloids [8]. Accordingly, it has a very high catalytic activity (each molecule of AChE degrades approximately 25000 molecules of acetylcholine (ACh) per second into choline and acetic acid [9]), which is inhibited by AFB1 [10]. Biosensors with AChE as the biorecognition component can detect toxic organophosphates in addition to carbamate pesticides, nerve agents, and numerous other common toxins [11].

A literature survey has shown several studies on biosensors for AFB1 detection with AChE as the biological component [12,13,14]. Arduini et al. [15] have designed an optical biosensor for AFB1 detection using acetylcholinesterase (AChE), which is inhibited by this toxin. The degree of inhibition was quantified by the Ellman’s spectrophotometric method, resulting in a detection limit of 10 µg L^−1^. Ben Rejeb et al. [14] have developed a bio-electrochemical biosensor for AFB1 detection in olive oil. The inhibition was quantified by an amperometric method with choline oxidase immobilized on a SPE and LOD achieved at 2 ppb.

The key to obtain a sensitive AFB1 biosensor is immobilizing AChE on a transducer. A natural biopolymer film located on the electrode surface protects the enzyme from exposure to any higher-molecular pollutants that may exist in the examined sample. In particular, in a suitable biosensor, the enzyme must hold its quaternary structure when placed close to the transducer, while the film should be thin [16].

Interest in using sodium alginate (SA) as a part of drug development has increased in the past two decades due to its satisfactory properties. Sodium alginate, recognized as an excellent biopolymer, is a linear polysaccharide extracted from natural seaweed, and consists of β-d-mannuronic acid (M) and α-l-guluronic acid (G) connected by 1,4-glycosidic bonds, with varied M/G ratios [17].

Alginate, a similar promising bio-adsorbent, was favored over other resources for its numerous attributes, including biodegradability, hydrophilicity, abundance, and the presence of sites for its carboxylation purposes [18]. In addition, it is non-toxic, biocompatible, and widely used in the pharmaceutical and food industries [19].

Sodium alginate-modified electrodes can provide a favorable microenvironment for enzymes, thus improving their stability and maintaining their bioactivity, as well as prolonging the storage time of the biofilm [20,21,22,23,24,25] and further promoting biological activity to enhance the sensitivity of a biosensor. To the best our knowledge, there are no reports on the application of the sodium alginate composite to prepare an AChE based biosensor for the impedimetric detection of AFB1.

Based on the above discussion, in this study, we developed a novel gold electrode modified with sodium alginate for the immobilization of AChE. An impedimetric biosensor to detect AFB1 was therefore obtained in the presence of the AChE substrate, acetylthiocholine (ATCh).

## 2. Results and Discussion

### 2.1. Optimization of the Amount of Sodium Alginate

Sodium alginate solutions of different concentrations were prepared. The optimum concentration of the biopolymer was obtained after testing the response of the modified electrode by cyclic voltammetry. We found that the peak current becomes the highest for 0.1 M sodium alginate and the voltammogram shape is well-defined. This concentration was fixed, and three different electrodes with 5, 15, and 20 µL of the biopolymer were tested. When the concentration of sodium alginate is higher, the current decreases (Figure 1). The blocking effect of the deposited layer can be attributed to two reasons: (i) The physical barrier of the biopolymer layer that prevents the access of [Fe(CN)_6_] ^3−/4−^ to the underlying gold electrode and (ii) the electrostatic repulsion charge–charge between the surface COO^−^ groups of the sodium alginate and the redox couple. Thus, 15 µL was chosen as the optimum volume of sodium alginate for the modified gold electrode.

### 2.2. Electrochemical Behavior of the Modified Electrode with Sodium Alginate Biopolymer

Cyclic voltammetry (CV) was used to obtain the electrochemical activity of the working electrodes prior to as well as after their modification with the biopolymer layer. Every phase of the biosensor development was examined through CV using the Fe(CN)_6_^3−/4−^ redox couple. The difference between anodic and cathodic highest potentials (E_p_ = E_pa_ − E_pc_) in addition to the intensity of the peaks (I_pa_ and I_pc_) can be associated with the electron transmission ability of the electrodes. As depicted in Figure 2A, combined oxidation and reduction peaks of the Fe(CN)_6_^3−/4−^ pair could be visibly identified prior to sodium alginate modification on the Au electrode (Ipa = 1.60 mA; curve a). When the sodium alginate layer was superficially deposited, a reduction in the redox current (I_pa_ = 0.760 mA) is perceived (curve b). This difference must be estimated because of the obstructive influence of the thick layer of sodium alginate and the negatively charged COO^−^ groups on the electrode surface, which might act as an electrostatic obstacle and decrease the electron transfer rate at the gold electrode surface [26]. These properties might be further cumulatively characterized through EIS measurements. The half-circle diameter in the impedance spectrum is equivalent to the electron-transfer resistance R_CT_. This resistance transduces the electron-transmission kinetics of the redox reaction at the electrode contact. The increase in the diameter of the semi-circle corresponds to the rise in the interfacial charge transfer resistance (R_CT_) [27]. The curve in Figure 2B.a represents the variation in R_CT_ for the bare Au electrode with R_CT1_ = 60 Ω. As illustrated in Figure 2B, after the deposition of the SA layer, R_CT2_ increases up to 1600 Ω (curve b), which is attributed to the formation of the thick biopolymer layer. The formation of the sodium alginate film severely hinders the interfacial charge transfer. It is worthy to note that these results are in accordance with those obtained from the CV measurements. The superficial coverage (θ) of the biopolymer film was as per the equation, θ = 1 − R_CT1_/R_CT2_, where R_CT1_ was the charge transfer resistance on the bare Au electrode and R_CT2_ was the charge transfer resistance on the Au electrode modified with the biopolymer film. Accordingly, θ was determined to be 60.5%.

### 2.3. SEM Analysis of SA Gel Bead

Field emission scanning electron microscopy (JSM 5100 from JEOL; JSM-SEM) images were obtained using a fully computer-controlled workstation. Figure 3A,B shows the morphology of the polymer film deposited on the gold electrode and coated with a thin gold layer applied by sputtering with thickness limited to 300 nm. Sodium alginate covered the entire surface, and the layer had a higher density and more wrinkles but low porosity.

### 2.4. Electrochemical Behavior of ATCh at AChE/SA/Au Electrode

The electrochemical behaviors of the AChE/SA/Au film electrode toward ATCh were investigated by CV with scan rate of 50 mV/s and EIS. Figure 4A shows the CVs of the AChE/SA/Au flexible film electrode without ATCh (curve a) and with 50 mg/mL ATCh (curve b) in 0.1 M phosphate buffer saline (PBS) (pH = 7.4). In the absence of ATCh, no redox peak appeared between 0.2 and 0.6 V, which confirms the stability of the AChE/SA/Au film electrode in the potential region. After adding 50 mg/mL ATCh solution, an oxidation peak at 0.49 V was obtained with a significant increase in peak current, indicating that the AChE/SA/Au film electrode exhibits electrocatalytic activity to ATCh. These results demonstrate that AChE was successfully immobilized on the SA/Au film electrode and the biopolymer did not influence the functionality of AChE, according to the following equations.
(1)$$\begin{array}{c} {\left(H_{3}C\right)}_{3}N^{+}CH_{2}CH_{2}SCOCH_{3}\\ Acethylthiocholine \end{array}\overset{AChE}{\to }\begin{array}{c} {\left(H_{3}C\right)}_{3}N^{+}CH_{2}SH\\ Thiocholine \end{array}+CH_{3}COOH$$
(2)$$2{\left(H_{3}C\right)}_{3}N^{+}CH_{2}CH_{2}SH\;\underset{Electrode}{\to }\;\begin{array}{c} S\\ |\\ S \end{array}\begin{array}{c} CH_{2}CH_{2}N^{+}{\left(CH_{3}\right)}_{3}\\ CH_{2}CH_{2}N^{+}{\left(CH_{3}\right)}_{3} \end{array}\begin{array}{c} +2H^{+}+2e^{-} \end{array}$$

### 2.5. Determination of the AChE Activity

First, it was necessary to determine an optimal concentration of ATCh as a substrate for subsequent inhibitory analysis. It is known that for biosensor analysis based on reversible inhibition, the working substrate concentration is often within a range corresponding to the linear part of a calibration curve of the biosensor used. The AChE activity was evaluated by measuring the product concentration of the enzymatic reaction [28]. Therefore, we studied the responses of the biosensor by recording the impedance spectra after the injection of acetylthiocholine in PBS solution (0.1 M, pH 7.0) (Figure 4B).

To obtain calibration curves, the values of ΔR_CT_ = R_CT_ − R_CT_ (0) were calculated, where R_CT_ (0) refers to R_CT_ for [ATCh] = 0. The curve was plotted vs. [ATCh]. Figure 5A,B reveals a linear measurement range for up to 4.95 mM ATCh. When the range was known, it was crucial to detect an ideal concentration of acetylthiocholine chloride as a substrate used for additional inhibitory examination. The biosensor exhibited the maximum sensitivity to aflatoxin B1 at the ATCh concentration of 0.09–4.95 mM. In the subsequent experiments, we used 4 mM ATCh as the substrate concentration, because the biosensor activity to this concentration was found to be greatly efficient, and its value is represented on the linear segment of the calibration curve.

### 2.6. Calibration of AFB1 Biosensor

For irreversible inhibitors, the enzyme–inhibitor interaction results in the formation of covalent bonds between the enzyme active center and the inhibitor. The term “irreversible” means that the decomposition of the enzyme-inhibitor complex results in the destruction of the enzyme, e.g., its hydrolysis oxidation. The procedure of inhibition is illustrated in Scheme 1.

If the inhibitor is not present in the system, ATCh would be transformed into thiocholine and acetic acid, as presented in Equation (1) If the inhibitor exists in the test solution, the concentration of thiocholine is completely diminished, i.e., thiocholine and acetic acid are not formed; in other words, it absolutely inhibits the conversion, as presented in Scheme 1. Under the influence of applied voltage, thiocholine is oxidized. The anodic oxidation current is inversely proportional to the concentration of toxic complexes in the sample and the time of contact.

The biosensor signal associated with the introduction of 4 mM acetylthiocholine through the experimental cell was determined, and its value was established as 100%. Subsequently, the aflatoxin B1 solution, whose concentration is to be determined, was added to the measurement cell [29]. The diameter of the Nyquist circle was found to increase with the addition of AFB1 (Figure 6A), indicating a rise in R_CT_. Using the Z_plot_/Z_view_ software, the value of R_CT_ was designed for each AFB1 concentration by fitting the experimental data.

The level of enzyme inhibition (*I%*) was determined by comparing the biosensor response to the substrate concentration before (*A*_0_) and after (*A_i_*) inhibition according to the Equation (3):
(3)$$I\%=(\frac{A_{0}-A_{i}}{A_{0}})\times 100\%$$

The inhibition of AChE activity (*I%*) proportional to AFB1 concentration could be measured assuming that there was a reduction in the degree of thiocholine production and AFB1 binding to the AChE site. Non-competitive inhibition probably proceeds during this period [30]. The entire measuring cycle lasted less than 10 min, representing an important advantage in rapid field testing.

The association between the inhibition percentage (*I%*) and the corresponding AFB1 concentration (fluctuating from 0.1 ng/mL to 100 ng/mL) is shown in Figure 6B. We perceived that with the increase in the concentration of Aflatoxin B1 from 0.1 ng/mL to 10 ng/mL, the inhibition increased linearly. The equation of the linear section of the inhibition curvature was y = 15.48 ln (x) +14.3 with a good correlation coefficient (R^2^ = 0.995). The calibration curve was plotted after an incubation period of 10 min. The inferior limit of the linear portion, defined as the concentration providing inhibition of 20%, was 0.1 ng/mL AFB1 (0.1 ppb). This value is less than the authorized limit value of the European Community regulation for human food (2 ppb) [16].

Finally, it was appropriate to compare the biosensor assay for aflatoxin detection, specifically by the established impedimetric AChE-modified biosensor, with current traditional techniques (Table 1). The frequently used techniques are TLC, HPLC, enzyme immunoassay (EIA), and their combinations. We correlated the key features, namely, the necessity for sample pretreatment, assay time, and the limit of aflatoxin detection. The primary benefits and disadvantages of the techniques are mentioned in Table 1.

The disposable impedimetric biosensor showed improvements in sensitivity and stability. Compared with the results obtained with ELISA, the immunosensor showed acceptable accuracy; it was also faster than HPLC and used less expensive reagents than the specific antibodies adopted in ELISA.

### 2.7. Reproductibility and Stability of the Biosensor

This test was performed to determine whether the biosensor signal decreased after introducing aflatoxin into the solution due to the inhibition of a bioselective component, and not because of unstable current and excessive fault in measurements. We observed active reproducibility, one of the best primary features of biosensors. The reproducibility of the biosensor was investigated by repeating the experiment with three different biosensors prepared in similar conditions. For the measurements, the biosensor was kept in a buffer solution at constant stirring and washed twice for 2 min. The established biosensor was categorized via reproducible signals detected by the direct resolution of the focal substrate beside the addition of aflatoxin B1, with a maximum error of around 8.5% [30]. This value indicates that our biosensor provides a good reproducibility of the fabrication protocol.

When the SA/AChE electrode was stored at 4 °C and then measured at intervals over several days, no obvious decrease of the current response was observed for 28 days of storage. After 45 days, the biosensor still retained 84% of the initial response. The superior stability of the SA/AChE electrode was attributed to the good film forming ability, the high mechanical strength, and the biocompatible environment of the sodium alginate biopolymer.

### 2.8. Specificity

The choice of the proposed impedimetric biosensor for detecting more groups of toxicants was also investigated. Figure 7 illustrates the experimental results of the established AChE-modified biosensor to AFB1, ochratoxin A (OTA), and aflatoxin M1. The biosensor signal was repressed by all groups of toxicants to varying extents. The highest sensitivity was achieved for aflatoxin B1 (57.9 kΩ/dec).

### 2.9. Rice Sample Analysis

The biosensor was used to detect AFB1 in rice. Impedance measurements were conducted with similar electrochemical probes (100 kHz, 10 mV). The changes in the signal were different between the spiked samples and blank samples, with a detection limit of 0.5 ng. mL^−1^ (0.5 ppb). The proposed method could detect AFB1 in spiked rice samples as low as 2 µg/kg, indicating that it is acceptable for AFB1 detection in spiked rice samples at the level of regulatory relevance.

## 3. Conclusions

A new impedimetric biosensor for determination of aflatoxin B1 through inhibition was developed. As a sensing bioelement, AChE was immobilized using sodium alginate natural biopolymer matrix on an Au electrode. The operational features of the AChE-biosensor for AFB1 analysis were considered and adjusted. The sensor obtained high sensitivity to aflatoxin B1 detection in a dynamic range from 0.1 to 10 ng/mL, with a detection limit as low as 0.1 ng/mL. This low detection limit is one of the most promising features of the developed impedimetric system for AFB1 detection in comparison with other analysis methods like spectrophotometric techniques.

The results from this study concluded that sodium alginate composite exhibited an improvement in the performance of the biosensor (storage stability, and reproducibility).

The ease of operation and cost-efficiency of the recommended method for AFB1 recognition, as well as the acceptable results found in terms of retrieval in actual samples (rice), justify the great potential of this test as a screening method for AFB1 recognition in actual samples.

Sodium alginate biopolymer providing a biocompatible host matrix that retained enzyme molecules by chemical cross-linking. It is expected that this simple and promising approach of biomolecule immobilization will be useful in the development of biosensors.

## 4. Materials and Methods

### 4.1. Chemicals

Glutaraldehyde (GAD, 25% *v*/*v*, aqueous solution), acetylcholinesterase (C 3389–500UN), aflatoxin B1, ochratoxin A, aflatoxin M1, acetylthiocholine chloride (ATCh), glycerol, bovine serum albumin, and sodium alginate (from brown algae, viscosity ≥ 2000 cP, 2% (25 °C) (lit.) were purchased from Sigma Aldrich.(Merck, Darmstadt, Germany), Aflatoxin B1 was solubilized in methanol (1 mM), followed by dissolution in water.

### 4.2. Instrumention

Electrochemical analyses, namely, cyclic voltammetry and impedance spectroscopy were performed using AutoLab (PGSTAT 302 N, Eco Chemie). The measurement set-up comprised a 3-electrode system. A platinum wire was used as the auxiliary electrode and a Hg/HgCl/KCl saturated electrode as reference electrode. The gold electrode was used as the working electrode. All electrochemical studies were performed in a dark Faraday cage at *T* = 296 ± 3 K (23 ± 3 °C)

### 4.3. Electrochemical Characterizations

In this study, the sodium alginate-modified electrode was characterized by cyclic voltammetry (CV). The potential was cycled from −400 mV and +600 mV (against SCE) at a scanning speed of 100 mV/s until numerous successive curves were overlaid. Phosphate buffer saline (PBS, 20 mM, pH 7.0) containing a 5 mM Fe[(CN)]^3–/4–^ couple was chosen as the electrolyte. Faradaic EIS characterization of the modified electrode was performed along with 20 mM PBS (pH 7.0), by applying a slight sinusoidal modulation (amplitude 10 mV; frequency varying from 100 MHz to 100 kHz). The excitation voltage of 10 mV was overlaid to the system at the open-circuit potential. Then, the Nyquist plots of the modified electrode were modeled in the Randles modified circuit, accounting for the presence of the film formed by the functionalization.

This electric circuit (Figure 8) is composed of the resistance in ohmic contacts (Rs), the charge transfer resistance (Rct) that transduces the charge transfer rate of the redox probe at the electrode surface, the imperfect double-layer capacitance between the electrode and the electrolyte (CPE), and the specific electrochemical element of diffusion Warburg impedance (Z_W_).

The determination of the different elements of the equivalent electrical circuit (CEE) was performed using the software for each registered Zview Nyquist diagram. Z scheme/Z view modeling software (Scriber and Associates, Charlottesville, NC, USA) was used to adjust the Faradaic impedance spectra

### 4.4. Development of the Sodium Alginate-Acetylcholinesterase-Based Biosensor

The Au electrodes (300-nm gold/30-nm titanium on a silicon substrate) were fabricated by the Laboratory of Analysis and Architecture of Systems (LAAS, Toulouse, France, member of the French RENATECH network) using standard silicon technologies. Prior to functionalization, the Au electrodes were sonicated for 10 min in acetone, dried under an N_2_ stream and then immersed in a piranha solution (H_2_O_2_:H_2_SO_4_ (3:7 *v*/*v*)) for 5 min at room temperature and finally washed with ethanol. After this step, the gold electrodes were washed thoroughly with ultrapure water and dried under N_2_ flow. Subsequently, the electrodes were modified with 15 µL sodium alginate dissolved in an acetate buffer solution (0.1 M).

### 4.5. Immobilization of AChE via GA Cross-Linking

AChE (5 mg; 500 UN) was added to BSA (5%, *w*/*v*) and glycerol (10%, *w*/*v*) in 20 mM phosphate buffer. This solution was thoroughly homogenized and allowed to stabilize at room temperature for 15 min. Subsequently, 20 µL of the homogenized mixture was deposited onto the modified gold electrode. Then, the biosensor was kept in soaked glutaraldehyde gas for 10 min for cross-linking and the final Au modified electrode was stored for 24 h at 4 °C.

### 4.6. Fabrication of the Impedimetric Biosensor

The entire impedimetric biosensor manufacturing process is presented in Scheme 2. The as-prepared biosensor was washed with distilled water prior to measurements to remove the excess unbound components on the membrane. The AChE biosensor works on the principle of inhibitory effects. In the AChE biosensor, the substrate, acetylthiocholine, is transformed into thiocholine and acetic acid. Thiocholine is oxidized via the functional voltage. In the presence of an inhibitor (AFB1), the conversion of acetylthiocholine declines [31]. Typical solutions of AFB1 were prepared in methanol, considered as the favored solvent for AFB1. The grade of inhibition was determined for increasing concentrations of AFB1. The variation in the electron-transfer resistance after AFB1 addition was used to evaluate the extent of inhibition. All measurements were performed in a minimum of three replicates.

### 4.7. Determination of AFB1 in Rice Samples

Non-contaminated rice (from a local market) was first ground in a household blender. Aliquots (1 g) of ground rice were spiked with AFB1 at different concentrations and mixed in a vortex mixer. After adding 5 mL of extraction solvent (80% methanol), the samples were mixed by shaking for 45 min and then centrifuged at 5000 rpm for 10 min. The supernatant was carefully removed and diluted with PBS (1–5 *v*/*v*).

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
