# Peer review of "An Acetylcholinesterase Inhibition-Based Biosensor for Aflatoxin B1 Detection Using Sodium Alginate as an Immobilization Matrix"

_toxins, 2020, doi:10.3390/toxins12030173_

Round 1

Reviewer 1 Report

I have carefully revised the new version of the manuscript and It has been significantly  improved. I only would recommend to improve figures quality before its publication in Toxins.

Author Response

Reviewer #1

Comment : I have carefully revised the new version of the manuscript and It has been significantly  improved. I only would recommend to improve figures quality before its publication in Toxins.

Response : All the figures were redrawn in order to improve their quality

Reviewer 2 Report

The article has been considerably improved, however it continue to be not novel. Moreover, the benefits of using sodium alginate continue to be not clear.

Other considerations:

- The SEM images should be in the same scale in order to be able to compare them.

- Data values about rice sample analysis must be added.

Author Response

Reviewer #2

Comment : The article has been considerably improved, however it continue to be not novel. Moreover, the benefits of using sodium alginate continue to be not clear.

Response : The rôle of alginate and the novelty of the paper is underligned in the introduction.

Sodium alginate-modified electrodes can provide a favorable microenvironment for enzymes, thus improving their stability and maintaining their bioactivity as well as prolonging the storage time of the biofilm [20–25] and further promotes biological activity to enhance the sensitivity of a biosensor. To the best our knowledge, there are no reports on the application of the sodium alginate composite to prepare an AChE based biosensor for the impedimetric detection of AFB1.

Moreover the good performance obtained  using sodium alginate is underligned in §2.7

When the SA/AChE electrode was stored at 4 0C and then measured at intervals over several days, no obvious decrease of the current response was observed for 28 days of storage. After 45 days the biosensor still retained 84% of the initial response. The superior stability of the SA/AChE electrode was attributed to the good film forming ability, the high mechanical strength and biocompatible environment of the sodium alginate biopolymer.

Other considerations:

- The SEM images should be in the same scale in order to be able to compare them.

SEM images were now obtained with the same scale.

- Data values about rice sample analysis must be added.

The data obtained in rice samples are given in §2.9 :

The changes in the signal were different between the spiked samples and blank samples, with a detection limit of 0.5 ng. mL–1 (0.5ppb). The proposed method could detect AFB1 in spiked rice samples as low as 2 µg/kg,

Round 2

Reviewer 2 Report

The data about rice sample analysis continue to be scarce. Nothing about the precision and accuracy of the measurements is said. Nothing is said about how many samples are tested and with what concentration are spiked.

This manuscript is a resubmission of an earlier submission. The following is a list of the peer review reports and author responses from that submission.

Round 1

Reviewer 1 Report

In this papers, authors present an electrochemical biosensors for aflatoxin B1 detection based on acetylcholinesterase inhibition using sodium alginate as immobilization matrix. Although the topic is so interesting and the manuscript is well organized, I think this paper needs substantial improvements. I suggest the paper for publication after major revision addressing the issues presented below:

-Authors should summarize in the introduction other biosensor for the aflatoxin B1 detection and remark the novelty of the new biosensor presented.

-Where is Figure1? All the Figure´s number are wrong, please correct this.

- In my opinion, Figures quality, in general, should be improved.

-Figures caption of Figure 2, 4 and 5 should be changed (For example: Figure 2. Cyclic voltammetry (A) and Nyquist diagrams (B) of Au  bare electrode (a) and Au electrode modified by sodium alginate (b) in the presence of 5 mM Fe(CN6)3- /4- 20 mM phosphate  buffer, pH 7.0.

Experimental details as frequency range and other details should be move to experimental section.

- Authors should explain in more detail Figure 4 and also include control experiments of Figure 4.This author request for the Au electrode modified with sodium alginate in PBS before and after AFB1 addition ( Cyclic voltammograms and Nyquist plots.

-Some misspellings must be corrected:

 In the abstract:  “biosensor assay “should be replaced by “biosensor”

In 2.1. and 4. Sections, formula of Fe (CN)6 3−/Fe (CN)64-  should be corrected as well as some strange symbols as 760?A or 1630?A

Reviewer 2 Report

In this paper, the authors developed an electrochemical biosensor for aflatoxin based on the inhibition of the enzyme acetylcholinesterase.  The enzyme was immobilized on a gold electrode modified with sodium alginate.

The manuscript is not well written. It is not novel and some information are incomplete or not well discussed. For example, the advantages of using sodium alginate are not explained or showed (with data). Other comments are:

Reference 3 should be the site of the International Agency for Research on Cancer or the document of this institution where is indicate that AFB1 is classifies as first category carcinogenic substance. The same for reference 5. The sentence “To obtain a sensitive AChE biosensor, the key is to immobilize AChE on the transducer” is not correct. The sensitivity of a biosensor depends on many other factors. Page 1, line 41: “film of the membrane”. Which membrane? It is not clear what are you referring to. Figure 1 is at the end of the manuscript and it should appear the first. Page 2, lines 71 and 73: there are a symbol incorrect (1630 ?A and 760 ?A). The imagens have very bad resolution. Section 2.2: it is “SEM analysis of SA gel bead” and it discuss more things that the SEM analysis. Page 6, line 149-150: “This value is not equivalent to the authorized limit of the European Community regulation for human food, which is 2 ppb”. Concentration valued should be indicate in the same units in order to comparing them. Table 1: time of what? The time of prepare the sensor is very long. It should be include in the Table. The sensor has not disadvantages? It is impossible since the perfect analytical method does not exist. The sample pretreatment should be specified. Reproducibility: 8.5% does not indicate “extremely reproducible” signals. Table 2: sample analysis for other concentrations (e.g. 0.25, 1 and 5 ng/mL) should be included. Electrode modification with alginate: it should be explained why 0.1 M of sodium alginate is used. Moreover, 10 μL as volume to modify should be also tested. Lines 252-264 should be included in the results section and not in “experimental materials and design”. In line 265 it is indicate that AFB1 is prepared in methanol and in Section 4.1 is indicated that it is prepared in ethanol (first dilution) and then in water.